# New Views of the DNA Repair Protein Ataxia–Telangiectasia Mutated in Central Neurons: Contribution in Synaptic Dysfunctions of Neurodevelopmental and Neurodegenerative Diseases

**DOI:** 10.3390/cells12172181

**Published:** 2023-08-30

**Authors:** Sabrina Briguglio, Clara Cambria, Elena Albizzati, Elena Marcello, Giovanni Provenzano, Angelisa Frasca, Flavia Antonucci

**Affiliations:** 1Department of Medical Biotechnology and Translational Medicine (BIOMETRA), University of Milan, Via F.lli Cervi 93, 20129 Segrate (MI) and via Vanvitelli 32, 20129 Milan, MI, Italy; sabrina.briguglio@unimi.it (S.B.); clara.cambria@unimi.it (C.C.); angelisa.frasca@unimi.it (A.F.); 2Division of Developmental Biology, Cincinnati Children’s Hospital Medical Center, Cincinnati, OH 45229, USA; elena.albizzati@gmail.com; 3Department of Pharmacological and Biomolecular Sciences, University of Milan, Via Giuseppe Balzaretti 9, 20133 Milan, MI, Italy; elena.marcello@unimi.it; 4Department of Cellular, Computational and Integrative Biology—CIBIO, University of Trento, Via Sommarive 9, 38068 Trento, TN, Italy; giovanni.provenzano@unitn.it; 5Institute of Neuroscience, IN-CNR, Via Raoul Follereau 3, 20854 Vedano al Lambro, MB, Italy

**Keywords:** ATM, hippocampus, synapse, GABA, glutamate, autism spectrum disorders, Alzheimer’s disease

## Abstract

Ataxia–Telangiectasia Mutated (ATM) is a serine/threonine protein kinase principally known to orchestrate DNA repair processes upon DNA double-strand breaks (DSBs). Mutations in the *Atm* gene lead to Ataxia–Telangiectasia (AT), a recessive disorder characterized by ataxic movements consequent to cerebellar atrophy or dysfunction, along with immune alterations, genomic instability, and predisposition to cancer. AT patients show variable phenotypes ranging from neurologic abnormalities and cognitive impairments to more recently described neuropsychiatric features pointing to symptoms hardly ascribable to the canonical functions of ATM in DNA damage response (DDR). Indeed, evidence suggests that cognitive abilities rely on the proper functioning of DSB machinery and specific synaptic changes in central neurons of ATM-deficient mice unveiled unexpected roles of ATM at the synapse. Thus, in the present review, upon a brief recall of DNA damage responses, we focus our attention on the role of ATM in neuronal physiology and pathology and we discuss recent findings showing structural and functional changes in hippocampal and cortical synapses of AT mouse models. Collectively, a deeper knowledge of ATM-dependent mechanisms in neurons is necessary not only for a better comprehension of AT neurological phenotypes, but also for a higher understanding of the pathological mechanisms in neurodevelopmental and degenerative disorders involving ATM dysfunctions.

## 1. Introduction

Ataxia–Telangiectasia Mutated (ATM) is a serine/threonine protein kinase whose function has been principally attributed to DNA damage response (DDR). Upon recruitment and activation by DNA double-strand breaks, it phosphorylates several key proteins, mostly tumor suppressors, that initiate activation of the DNA damage checkpoint, leading to cell cycle arrest, DNA repair, and apoptosis [1]. Furthermore, mutations in the *Atm* (AT mutated) gene are associated to the neurodegenerative condition Ataxia–Telangiectasia (AT). AT is a recessive disorder with an extremely variable phenotype, characterized by uncoordinated and ataxic movements as a result of cerebellar atrophy or dysfunction [2], together with immune defects, genomic instability, and predisposition to cancer [2]. AT patients show variable phenotypes, ranging from neurologic abnormalities to cognitive impairments, which can appear early and are thought to be related with the progression of disease. In particular, AT patients displaying ataxic symptoms early in life show cognitive deficits reflecting cerebellar neuropathology, whereas patients with late symptoms onset show cognitive impairments reflecting noncerebellar pathology [3], thus suggesting additional ATM functions in different brain regions. Accordingly, neuropsychiatric features recently recognized in AT patients [4] appear unrelated to the canonical functions of ATM in the field of DDR and new ATM activities have been largely described in adult neurons, indicating the need of a complete understanding of ATM functions. The current hypothesis is that ATM deficiency is likely responsible for different pathological features of AT through distinct effects occurring in specific brain regions and in diverse cell populations [5,6,7]. Indeed, ATM regulates neuronal communication as indicated by changes in hippocampal and cortical synapses of *Atm*-deficient mice. Since synapses are essential units of neuronal physiology responsible for all body functions and aspects of cognition, recent data pinpoint a role of this kinase in brain pathologies such as neurodevelopmental and neurodegenerative diseases, besides AT. 

## 2. The DNA Damage Response

The primary role of DNA repair proteins is essential throughout cell life to guarantee genome stability and cell survival; indeed, they orchestrate a complex cellular response to repair damaged DNA by detecting DNA lesions and transducing signaling cascades. Notably, DNA damage is highly frequent and generated by both endogenous and exogenous genotoxic agents: reactive oxygen species (ROS), produced during normal cellular metabolism, can oxidize DNA bases, leading to single-strand breaks (SSBs); the ultraviolet component of sunlight may generate clustered DNA lesions and bulky adducts and chemicals and antitumor agents may be responsible for the generation of double-strand breaks (DSBs), possibly resulting in chromosome breaks and genomic instability during cell division. Moreover, DNA replication phase has high mutagenic potential upon nucleotide misincorporation. Thus, to guarantee the necessary DNA integrity, cells have developed multiple DDR mechanisms [8]. SSBs are repaired through a series of mechanisms termed base excision repair (BER) [9]; bulkier single-strand lesions that distort the helical structure of DNA are processed by nucleotide excision repair (NER) [10,11]; incorrectly paired nucleotides are recognized and corrected by the mismatch repair (MMR) pathway [12,13]. DSBs, which represent the most deleterious conditions leading to possible chromosome breaks and altered genomic integrity, are coped with homologous recombination (HR) or non-homologous end joining (NHEJ), according to the stage of the cell cycle. In proliferating cells, HR, which requires the sister chromatid as a repair template, is the process mainly responsible for repairing DSBs. Here, mitotic cells stall cell cycle progression to repair DNA damage before potential faults pass to the next replicative generation. However, if the damage is too extended to be fixed, the cell has two main potential fates: senescence, characterized by permanent cell cycle arrest, or apoptosis, the regulated program of cell death [14]. In the case of post-mitotic, terminally differentiated cells such as neurons, repair mechanisms do not rely on the cell cycle and restoration of DSBs occurs through NHEJ, which is active throughout the entire lives of cells. Therefore, in post-mitotic cells, the activation of different responses is critical in preventing the accumulation of unrepaired DNA damage which may induce (i) mutagenesis, such as base substitutions, small insertions/deletions, and gross chromosomal rearrangements which could result in altered cellular phenotype, dysfunctions, and diseases or (ii) cell death as found in pathologies characterized by tissue atrophy, i.e., neurodegenerative diseases. In addition, the accumulation of DNA damage is linked mechanistically to other disease processes implicated in both neurodegenerative and developmental disorders such as mitochondrial alterations, redox dysregulation, and defects in protein quality control [15,16,17,18]. Accordingly, genetic mutations in proteins that operate in BER/SSB repair (BER/SSBR) pathways such as TDP1, APTX, and PNKP have been directly linked to neurological disorders. Importantly, expression levels of many NHEJ and BER proteins (which operate during all stages of the cell cycle) have been found to decline with age in mouse and rat brains, reducing repair efficiency and culminating in neurological disorders [19,20,21]; on the contrary, expression of HR proteins persists postnatally, during development and in adult life of differentiated cells such as neurons. Indeed, even though ATM is a serine/threonine protein kinase which acts as a central regulator of the DDR in all eukaryotic cells, in the last decade, increasing evidence has demonstrated a wide expression of ATM also in non-proliferating neurons where it exerts different functions only marginally related to DNA repair. 

## 3. Role of DNA Double-Strand Break (DSB) Formation in Neuronal Physiology

Suberbielle E et al. and Madabhushi R et al. demonstrated that activity-dependent cells generate DSBs in neurons [22,23] and, in particular, that they occur in the promoters of several early response genes essential for learning and memory processes [23]. Thus, synaptic activity-induced transcription is associated with the induction of transient DSBs in the promoters of stimulus-inducible genes, suggesting an association between DSB generation and cognitive performances. The occurrence of DSBs in specific promoter regions is still under investigation, to understand if it (i) facilitates chromatin plasticity required to sustain gene transcription or (ii) poses a further threat to genome stability. Several studies reported that DSBs are necessary for gene expression. For instance, in 2006, Ju et al. reported that DSB generation by the TOP2β–PARP1 complex in the pS2 promoter is required for estrogen-induced transcription of the target gene in MCF7 cells [24]. In 2010, ATM has been found to repress gene expression from a distant promoter, thus defining a novel role for this kinase in influencing chromatin dynamics within euchromatic environments [25]. More recently, the chromatin modifier NuA4 complex, crucial for DNA repair processes [26], has been found to assemble in activated neurons around the inducible, neuronal-specific transcription factor NPAS4 to bind recurrently damaged regulatory elements and to recruit additional DNA repair machinery to stimulate their repair. Disruption of the NPAS4–NuA4 complex or its function leads to (i) dysregulation of activity-dependent gene expression, (ii) increased DNA breaks in activity-regulated promoters and enhancers, (iii) impaired localization of protective repair machinery, and (iv) reduced lifespan as seen in Npas4 knockout mice [27]. Thus, the formation of activity-dependent DNA breaks is a transient and essential process by which chromatin remodeling and gene transcription occur during intense activity and are subsequently resolved by activation of NPAS4–NuA4.

## 4. Role of ATM in Synaptic Physiology

In neurons, activation of specific DNA repair proteins is necessary for DNA damage processes, chromatin remodeling, and gene transcription; also, intense research activities have demonstrated a clear contribution of such proteins in the proper functionality of synapses, as in the case of ATM.

The first piece of evidence for a possible role of ATM at the synapses of cerebellar granule cells was its direct interaction with b2-adaptin, one of the components of the AP2 adaptor complex. Therefore, it has been suggested that ATM can participate in the clathrin-mediated endocytosis of receptors [28] and the transport of vesicles from the trans-Golgi network to endosomes and lysosomes. In the same study, the authors displayed the ability of ATM to interact in vitro with β-NAP, a neuronal-specific beta-adaptin homolog, identified in a patient with cerebellar degeneration [28]. The study clearly demonstrated an extra-nuclear role of ATM, hypothesizing novel DDR-unrelated functions for the kinase and suggesting new mechanisms for cerebellar neurodegeneration. However, the roles of ATM in extra-cerebellar regions remained unexplored. Many years later, in the laboratory of Karl Herrup, Jiali Li demonstrated the cytoplasmic distribution of ATM in cortical neurons and, in particular, at synapses, showing its colocalization with two synaptic vesicle proteins, VAMP2 and synapsin-I, both of which must be phosphorylated to bind ATM [29]. Thus, starting with this initial evidence, the same authors hypothesized a defective non-nuclear function of ATM, unrelated to DNA repair processes, at the basis of neurological symptoms of AT. Cortical *Atm*-depleted neurons displayed a reduced rate of spontaneous vesicular dye release, confirming the great importance of ATM in neuronal functions and in synaptic vesicle recycling [29]. A further direct contribution of ATM to presynaptic vesicle physiology was confirmed by (i) a three-dimensional reconstruction revealing ATM as significantly more closely associated with the presynaptic marker Piccolo than with the post-synaptic marker Homer1 and (ii) electrophysiological results of short-term plasticity (STP) in hippocampal slices of *Atm* KO mice, indicating a reduced facilitation [30]. Furthermore, a clear impairment in the induction of post-synaptic plasticity in *Atm* KO CA1 hippocampal neurons by long-term potentiation (LTP) experiments was report, highlighting ATM as a new player in the establishment of synaptic plasticity. Overall, these data lead to the hypothesis that cognitive impairments associated with AT are not exclusively generated by cerebellar degeneration, as generally believed [31], but may also arise from a direct role of ATM in synaptic plasticity. Furthermore, these results indicate the presence of different mechanisms underlying the AT neurological phenotypes in addition to those involved in DNA damage.

### 4.1. Role of ATM in the Excitatory-to-Inhibitory Switch of GABA and Related Hippocampal Synaptic Dysfunctions

Starting with the evidence that ATM participates in the correct functioning of the central synapse and taking into consideration that AT patients display specific cognitive disabilities [3], the eventual contribution of ATM in the balance between excitation and inhibition (E/I) has been investigated. Accordingly, impairments in E/I balance underlie several brain disorders. Circuitry formation, activity-dependent tuning of neuronal networks, and neuronal plasticity are brain properties controlled by the correct equilibrium between excitation and inhibition [32,33,34,35,36,37,38,39]. In adult brains, excitatory (depolarizing) and inhibitory (hyperpolarizing) inputs are driven by the release of glutamate and GABA neurotransmitters, respectively. However, early in development, GABAergic synaptic transmission exerts an excitatory action, being responsible for neuronal depolarization [40,41]. This GABA-mediated depolarizing/excitatory action is consequent to developmental changes in chloride homeostasis, tightly regulated by two chloride cotransporters called NKCC1 and KCC2. The chloride importer NKCC1 is highly expressed in immature neurons and maintains high intracellular chloride concentrations; thus, activation of GABA-A receptors results in chloride efflux and membrane depolarization. During development, the expression of the chloride exporter KCC2 is up-regulated, with concomitant downregulation of NKCC1, and this shift results in a progressive decrease in intracellular chloride concentration, leading to the excitation-to-inhibition switch in GABA-A receptor-mediated signaling [42]. GABAergic responses undergo a permanent switch from being excitatory to inhibitory during the first postnatal period [43] and this process directly reflects the expression and activity of NKCC1 and KCC2. Of note, changes in this process may arise from altered levels of these two cotransporters, leading to an unbalanced E/I ratio and to brain pathologies. In this context, the electrophysiological analysis of excitatory (EPSC) and inhibitory post-synaptic currents (IPSC) in miniature and immunofluorescence analysis of density of glutamatergic and GABAergic terminals have demonstrated that 50% reduction in ATM levels (i.e., *Atm* heterozygosity) produces an excitatory/inhibitory unbalance towards inhibition in hippocampal neuronal cultures [44]. Interestingly, increased expression of the chloride extruder KCC2 in (i) *Atm* heterozygous neurons [44], (ii) neurons in which ATM expression was acutely down-regulated by a specific siRNA [44], and (iii) neurons treated with the ATM kinase activity inhibitor KU55933 [45] was demonstrated. Coherently with the high KCC2 expression level, a precocious development of the inhibitory system (i.e., excitatory-to-inhibitory GABA switch) was also highlighted by calcium imaging experiments. These rearrangements result from a process that involves the extracellular signal-regulated kinase 1/2 (ERK 1/2) and activation of the transcription factor Egr4 [45] which pushes KCC2 transcription during early postnatal development [46]. Therefore, in the absence of ATM, the higher Egr4 activity results in increased KCC2 levels which lead to an anticipated development of GABAergic system and an increased number of inhibitory synapses. Interestingly, the delivery of the KCC2 pharmacological blocker VU0240551 in *Atm* heterozygous developing neurons normalizes these two processes [45]. This occurs specifically in hippocampal structure, since normal GABA switching and KCC2 levels have been detected in cortical cells and lysates of *Atm* KO embryos and mice [47], suggesting a specific role of ATM in the hippocampus. A deeper description of synapse composition also unveiled increased levels of Gluk-1 and Gluk-5 subunit-containing kainate receptors (KARs) in *Atm*-depleted hippocampal cultures and tissues, in line with the evidence that KARs coexist with KCC2 in a macromolecular complex [48]. Of note, KCC2 activity is essential for KAR accumulation since treatment with the KCC2 blocker VU0240551 restored the increased expression of KARs in *Atm* heterozygous neurons during development [47]. Thus, early in life, ATM regulates KCC2 levels in the hippocampus by tuning Egr4 activity that, in turn, results in premature GABA switching, increased number of GABAergic synapses, and accumulation of KARs. Electrophysiological results displayed that KARs are fully active and responsible for a strong excitatory current evoked by electrical stimulation. Furthermore, analysis of post-synaptic fractions and confocal investigations displayed a specific presynaptic KAR accumulation wherein they promote clustering of glutamatergic synaptic vesicles in the Readily Releasable Pool (RRP, [47]), according to the KAR-metabotropic action [49,50]. Interestingly, the KCC2 pharmacological blocker VU0240551 was able to normalize glutamatergic vesicles in the RRP in *Atm* heterozygous neurons, in addition to restoring the expression of KARs [47] (Figure 1).

Collectively, these results indicate that ATM controls KCC2 levels during development and, thereby, regulates (i) the switching of GABA, (ii) E/I synaptic ratio, and (iii) presynaptic expression of KARs, essential for the correct development of both inhibitory and excitatory synaptic structure and neurotransmission. Considering that changes in KARs levels were undetectable in the cortex of *Atm*-deficient mice, where no variations in KCC2 expression and GABA switching were reported, these mechanisms occur exclusively in the hippocampus during development and are KCC2-dependent [47]. 

### 4.2. ATM and E/I Imbalance in Cortical Neurons

A similar E/I imbalance characterized by increased inhibition has been described in cortical neurons collected from *Atm* heterozygous and homozygous embryos by immunofluorescence analysis of excitatory and inhibitory synaptic puncta [51]. Super-resolution microscopy and coimmunoprecipitation analysis displayed the exclusive association of ATM with the vesicular transporter for glutamate VGLUT1 [51], confirming results collected in hippocampal cells and brain slices, whereas analysis of the endocytic pathway dynamics in synapses by FM4-64 dye uptake under selective treatment with ATM inhibitor KU-60019 revealed a deficiency in the endocytic process [51]. Thus, in cortical neurons, synaptic vesicle endocytosis, but not release, is an ATM-dependent process. Interestingly, electrophysiological recordings of excitatory and inhibitory post-synaptic currents (mEPSCs and mIPSCs) performed in *Atm*^+/−^ and *Atm*^−/−^ cortical neurons showed no change in the EPSC or IPSC frequency [47], pointing to a quite normal release of glutamate and GABA in *Atm*-deficient cortical cultures. Moreover, considering that normal GABA switching and KCC2 levels have been detected in cortical preparations [47], it is unlikely that E/I imbalance and the higher inhibition found in cortical *Atm* homozygous cultures result from a modification in the chloride homeostasis and/or in the development of GABAergic system, as in the hippocampus. Thus, ATM plays specific roles and supports selective functions in cortical and hippocampal structures, suggesting a brain area specificity. 

Overall, these results reveal that ATM is part of the cellular infrastructure that maintains the E/I balance of central neurons, even if it is implicated in distinct molecular pathways in the cortex and in the hippocampus. Different approaches have enabled the identification of novel functions for ATM in synaptic transmission and brain development, leading to a better understanding of the cognitive defects occurring in AT and, potentially, in a wide range of neuropathological conditions.

## 5. DNA Repair Proteins in Autism Spectrum Disorders (ASD)

Autism spectrum disorders (ASD) include a group of heterogeneous neurodevelopmental conditions, affecting ~1% of the population, which are diagnosed merely on the basis of core symptoms such as early-onset impaired social communication and interaction and restricted repetitive behaviors. Many other comorbidities are present as well, spanning from intellectual disability (in approximately 70% of cases), epilepsy, anxiety, sleeping disorders, and gastrointestinal issues [52,53]. 

While in some cases the specific cause of ASD is still elusive, there are some syndromes commonly associated with ASD that rely on specific genetic alterations. They include Fragile X (FXS), Cdkl5 Deficiency disorder (CDD), Cornelia de Lange syndrome (CDLS), and Tuberous Sclerosis Complex (TSC). Rett syndrome (RTT) is no longer classified as an autism spectrum disorder, although RTT patients often manifest autistic behaviors. Interestingly, several genetic mutations responsible for ASD syndromes have been identified and many of these genes codify for proteins essential for neuronal differentiation, development, and synaptic transmission. Importantly, in the last several years, a number of pieces of evidence have highlighted the involvement of non-neuronal pathways in neurodevelopmental diseases and ASD animal models, paving the way to the study of immune system cells and DNA damages (Table 1). Regarding the contribution of the immune system, clear dysfunctions have been described in the prenatal and postnatal periods. During gestation, prenatal insults including maternal infection and subsequent immunological activation may increase the risk of ASD in children. The postnatal environment, on the other hand, is characterized by related but distinct profiles of immune dysregulation, inflammation, and endogenous autoantibodies that persist within the affected individual [54]. In addition, increased levels of oxidative stress and DNA damage have been reported in ASD subjects as well as in the cerebellum of animal models for ASD [55,56,57,58]. Furthermore, transcriptomic analysis performed in lymphocytes from ASD individuals and in those exposed to γ irradiation revealed that several genes associated with DNA repair were down-regulated, thus suggesting a possible increase in oxidative DNA damage in lymphocytes from ASD individuals [58]. It has been shown that neurons display changes in DNA damage and in DSB formation. Recent studies identified recurrent DSB clusters (RDCs) in genes of mouse neural precursor cells (NPCs); most of these clusters occur in long neural-specific genes associated with neuropsychiatric diseases and cancers, suggesting potential impacts of DNA damage on neural development and function [59,60]. Moreover, growing evidence indicates that NPCs derived from ASD patients with macrocephaly undergo rapid cell cycle progression [61,62] and, recently, it has been demonstrated that NPCs derived from iPSCs reprogrammed from fibroblasts of ASD individuals with macrocephaly exhibit chronic DNA damage compared with NPCs derived from control subjects [63]. Thus, genome instability in NPCs may contribute to neurodevelopmental disorders such as ASD. Finally, preclinical studies in both genetic and pharmacological animal models for the study of autistic-like behaviors displayed increased expression of DNA repair kinases [64] along with that of topoisomerase IIβ (TOP2β), which is expressed in differentiation cells and neurons [65,66] and facilitates the expression of long-transcripts linked to autism [67].

### Contribution of ATM in Synaptic Dysfunctions of Neurodevelopmental Disorders

According to the Simons Foundation Autism Research Initiative (SFARI) database, more than 1000 candidate genes and copy number variation (CNV) loci have been associated with ASD. Interestingly, many of the identified genes, including NRXN, NLGN, SHANK3, UBE3A TSC1/2, FMR1, and MECP2, codify for proteins with an established role in synapse formation, maintenance, and functioning. In good accordance, the presence of synaptic defects in ASD is largely acknowledged, caused by abnormalities in synaptogenesis, synaptic maturation, elimination, functioning, and plasticity, leading to the definition of ASD as synaptopathies [68,69]. While excessive synapse formation and higher density of dendritic spines have been reported in autism and FXS, RTT and CDD are characterized by a reduced number of synapses. In general, synaptic defects represent a hallmark of ASD and are considered primary targets in ASD therapies. Structural alterations are reflected also in abnormal synaptic functionality. 

Specifically, studies conducted on both ASD animal models and patients, by clinical neuroimaging, and analysis conducted on iPSCs cells [70,71,72] converged on a disruption of the E/I balance both in cortical and sub-cortical regions. Alteration in the E/I balance emerges from increased glutamatergic activity along with a decrease in GABAergic transmission, whereas, at the molecular level, a reduction in KCC2 by impairing GABAergic maturation is considered the leading cause of this defect. It is peculiar that altered expression of KCC2 is commonly detected in several neurodevelopmental disorders. For instance, KCC2 levels are low in the brains of *Mecp2* KO mice, modeling RTT, and in post-mortem brain tissues and cerebrospinal fluid of RTT patients [73,74]. The relevance of KCC2 impairment for RTT is underlined by the evidence that increasing its expression by pharmacological compounds rescues cellular and behavior alterations [75,76]. Similarly, KCC2 down-regulation is observed in TSC patients concomitantly with an NKCC1 upregulation and in prenatally treated valproic acid (VPA) animals, modeling autism.

To our knowledge, although many data point to the involvement of ATM in the regulation of neuronal functions and in E/I balance, only one paper supports its role in the synaptic alterations of ASD so far. In detail, Pizzamiglio and collaborators recently demonstrated that ATM protein levels are increased in the hippocampus of both *Mecp2* KO mice and a VPA-induced autistic mouse model and that ATM inhibition by KU55933 is effective in normalizing KCC2 expression and rescuing abnormal E/I balance together with neuronal hyperexcitability [45]. Little evidence has further indicated the possible involvement of ATM also in CDD, reporting a reduction in ATM phosphorylation in *Cdkl5* KO animals [77] and an altered expression of factors associated with DDR, such as γH2AX and XRCC5 [78] which paralleled synaptic alterations. However, to date, no data have demonstrated the causative role of ATM for synaptic defects in other ASD models.

**Table 1 cells-12-02181-t001:** Literature review of ATM involvement in neurodevelopmental issues, sorted by process.

Process	Paper Title	Year	Ref.
Apoptosis	Requirement for Atm in Ionizing Radiation-Induced Cell Death in the Developing Central Nervous System	1998	[7]
	Defective neurogenesis resulting from DNA ligase IV deficiency requires Atm	2000	[5]
	DNA-damage response, survival and differentiation in vitro of a human neural stem cell line in relation to ATM expression	2009	[79]
	Alteration in 5-hydroxymethylcytosine-mediated epigenetic regulation leads to Purkinje cell vulnerability in ATM deficiency	2015	[80]
Autophagy	ATM-deficient neural precursors develop senescence phenotype with disturbances in autophagy	2020	[81]
Cell cycle	p38 MAPK-Mediated Bmi-1 Down-Regulation and Defective Proliferation in ATM-Deficient Neural Stem Cells Can Be Restored by Akt Activation	2011	[82]
	Alteration in 5-hydroxymethylcytosine-mediated epigenetic regulation leads to Purkinje cell vulnerability in ATM deficiency	2015	[80]
	Loss of Neuronal Cell Cycle Control in Ataxia Telangiectasia: A Unified Disease Mechanism	2005	[83]
Differentiation	DNA-damage response, survival and differentiation in vitro of a human neural stem cell line in relation to ATM expression	2009	[79]
DSBs	Requirement for Atm in Ionizing Radiation-Induced Cell Death in the Developing Central Nervous System	1998	[7]
	Defective neurogenesis resulting from DNA ligase IV deficiency requires Atm	2000	[5]
	DNA-damage response, survival and differentiation in vitro of a human neural stem cell line in relation to ATM expression	2009	[79]
	Alteration in 5-hydroxymethylcytosine-mediated epigenetic regulation leads to Purkinje cell vulnerability in ATM deficiency	2015	[80]
Epigenetic regulation	p38 MAPK-Mediated Bmi-1 Down-Regulation and Defective Proliferation in ATM-Deficient Neural Stem Cells Can Be Restored by Akt Activation	2011	[82]
	Alteration in 5-hydroxymethylcytosine-mediated epigenetic regulation leads to Purkinje cell vulnerability in ATM deficiency	2015	[80]
E/I balance	New Role of ATM in Controlling GABAergic Tone During Development	2016	[44]
	ATM and ATR play complementary roles in the behavior of excitatory and inhibitory vesicle populations	2017	[51]
	The DNA repair protein ATM as a target in autism spectrum disorder	2021	[45]
	ATM rules neurodevelopment and glutamatergic transmission in the hippocampus but not in the cortex	2022	[47]
Inflammation/ Immune response	ATM-deficient neural precursors develop senescence phenotype with disturbances in autophagy	2020	[81]
Mitochondrial activity	ATM-deficient neural precursors develop senescence phenotype with disturbances in autophagy	2020	[81]
Oxidative stress	Oxidative Stress Is Responsible for Deficient Survival and Dendritogenesis in Purkinje Neurons from Ataxia Telangiectasia Mutated Mutant Mice	2003	[84]
	DNA-damage response, survival and differentiation in vitro of a human neural stem cell line in relation to ATM expression	2009	[79]
	ATM is activated by ATP depletion and modulates mitochondrial function through NRF1	2019	[85]
	p38 MAPK-Mediated Bmi-1 Down-Regulation and Defective Proliferation in ATM-Deficient Neural Stem Cells Can Be Restored by Akt Activation	2011	[82]
	ATM phosphorylation of the actin-binding protein drebrin controls oxidation stress-resistance in mammalian neurons and C. elegans	2019	[86]
	ATM-deficient neural precursors develop senescence phenotype with disturbances in autophagy	2020	[81]

## 6. DNA Repair Proteins in Alzheimer’s Disease (AD)

The rise in life expectancy during the 20th century has resulted in a burgeoning number of individuals at high risk of neurodegenerative disorders. As the most prevalent type of dementia, Alzheimer’s disease (AD) is one of the most common diseases in elderlies, thereby representing a global health crisis [87]. AD clinical symptoms include progressive memory impairment, disordered cognitive function, and attendant psychotic symptoms. From a histological point of view, AD is characterized by two pathological hallmarks: the extracellular senile plaques, composed of β-amyloid (Aβ) peptide, and the intracellular neurofibrillary tangles of hyperphosphorylated tau protein [88]. The amyloid hypothesis has guided the development of potential treatments in the last 25 years [89]. Importantly, synaptic dysfunction and neuronal loss occur in the cortex and hippocampus from the very early stages of the pathology [88]. In AD pathogenesis, synaptic degeneration is central to the disease and may serve as a driving force, rather than a by-product, of AD pathology that leads to memory impairment. Dendritic spine loss is seen in post-mortem brains from AD patients and in the brains of AD mice models [90]. Substantial data indicate Aβ soluble oligomers as the primary influence driving synaptic dysfunction and spine loss [91]. The lack of synaptic contacts represents an early insult that advances with the disease and in AD patients, cognitive decline has a stronger correlation to synapse and dendrite loss than to neurofibrillary tangles or neuronal death [92]. Therefore, attempts to decipher the subtle alterations in synaptic function that underlie the earliest cognitive features of AD are important in identifying the mechanisms implicated in AD pathogenesis.

In this frame, DDR can be a relevant pathway in AD because of its role in neurodegeneration. Indeed, DNA damage has been found to activate a cell death program in terminally differentiated post-mitotic neurons and, thereby, has been implicated in the pathology of brain injuries and neurodegenerative conditions (Table 2) [93,94] and defects in DNA repair mechanisms frequently occur together with neurodegeneration. [95,96,97,98]. In particular, a direct involvement of DNA repair proteins has been strongly suggested [95,96] and a correlation between accumulation of DNA damage and cognitive decline has been proposed [97,99]. 

Unfortunately, the mechanisms leading DNA repair kinases to be directly involved in AD pathogenesis are still unclear. Most studies have investigated proteins of the BER pathway, especially the enzyme 8-oxoguanine glycosylase 1 (OGG1), in mild cognitive impairment subjects and sporadic AD patients. OGG1 is the protein that catalyzes the incision of DNA at sites of the oxidation product 8-hydroxy-2′-deoxyguanosine (8-OHdG) and its activity has been found consistently reduced in clinical studies [95,98,100]. The role of another key enzyme of the BER pathway, DNA polymerase beta (POLB), has also been investigated in the pathogenesis of AD [101]. POLB is loaded into neuronal DNA replication forks in response to Aβ treatment. The analysis of autoptic brain specimens of AD patients and control subjects revealed a progressive increase in POLB expression along with the severity of the pathology. However, POLB expression declines in AD patients at later stages of the disease, likely reflecting severe neuronal loss [101]. In addition to POLB, increased APEX1 (APE1/REF-1) staining in AD brain sections of the hippocampus and temporal cortex has been described [102,103], where APEX1 is also part of the BER pathway and functions as 5′-apurinic/apyrimidinic endonuclease. 

Considering the proteins belonging to the NER pathway, biochemical analysis revealed significantly increased levels of ERCC2 and ERCC3 in post-mortem brain tissues of AD patients at the early stages of the disease compared to control subjects [104].

Finally, proteins involved in the repair of DSBs have also been described to be altered in AD. In particular, reduced levels of the DNA-dependent protein kinase (DNA-PK) have been measured in AD lysates [105]. Also, reduction in MRN complex proteins (MRE11, RAD50, and NBN) occurs in AD individuals, specifically in the cortex, one of the brain regions most affected in AD. 

AD transgenic mice, which recapitulate key aspects of pathology, have increased neuronal DSBs at baseline and more severe and prolonged DSBs after physiological experiences such as explorative behavior, confirming the strict relation between synaptic plasticity and DSBs. Interestingly, administration of levetiracetam suppresses aberrant neuronal activity, improves learning and memory, and normalizes levels of DSBs [22].

In vitro, Aβ inhibits DNA-PK-dependent NHEJ and contributes to the accumulation of DSBs in PC12 cells [106]. Remarkably, blocking the extrasynaptic NMDA-type glutamate receptors prevents Aβ-induced DSBs in neuronal cultures [22], suggesting that DSB generation is dependent on the effect of Aβ on glutamatergic neurotransmission. 

Overall, it is clear that increased DSB formation together with low DNA repair efficacy represent a toxic condition with negative impacts on (i) genome stability [107], (ii) neuronal performances, and (iii) survival, as demonstrated in AD [107,108,109]. 

### Contribution of ATM in AD and Synaptic Dysfunctions of AD

The involvement of ATM in AD pathogenesis has been supported by data obtained from autoptic specimens of AD patients and AD mouse models. Indeed, ATM protein levels are altered in the lysates of human frontal cortex and cerebellum of AD patients. Results indicate a reduction in ATM signaling in the frontal cortex, whereas increased ATM levels were reported in the cerebellum of AD patients, pointing to specific roles of this kinase in brain areas diversely affected by the pathology [110]. In AD mice, evidence for loss of function of ATM was detected in three different AD transgenic mouse models [110], suggesting that alterations in ATM signaling are implicated in AD pathogenesis. 

In glial cells of mutant ATM Drosophila melanogaster flies, ATM impairment or deficiency may trigger innate immune responses that can contribute to neurodegeneration [111]. The histology of microglial cells in *Atm* KO mice is abnormal, and astrocytes from *Atm* KO mice show significant expressions of oxidative and endoplasmic reticulum stress and a senescence-like reaction [112,113]. Notably, *Atm* deficiency may disturb DNA repair and accelerate aging and neuroinflammation.

Thus, clinical and preclinical evidence supports the assumption that ATM may represent a new player contributing to neurodegeneration and synaptic dysfunction in AD. 

Concerning neurodegeneration, ATM signaling is implicated in cell death pathways and is necessary for apoptosis of post-mitotic neurons triggered by genotoxic or oxidative stress [7,114]. ATM is also a mediator of NMDAR-associated excitotoxicity and cell death. Primary neuronal cultures lacking ATM are partially protected from NMDAR-induced death. Furthermore, NMDAR-dependent increased caspase activity is abolished in cells lacking ATM [115]. Interestingly, the role of ATM inhibition as a potential therapeutic approach to counteracting neurodegeneration has been explored in Huntington’s disease (HD), where ATM genetic deficiency and the pharmacological inhibition achieved by KU-60019 are associated with an amelioration of pathological features [94]. Aβ can also induce oxidative DNA damage in in vitro models of AD [116,117] and blockage of ATM abrogates Aβ-induced apoptosis, further supporting the involvement of the DNA damage response in Aβ-mediated neurotoxicity [114]. Therefore, ATM deficiency increases the resilience of neurons to insults that normally elicit robust neuronal cell death. In line with this, the down-regulation of ATM signaling in AD patients and AD mice may be an attempt made by neuronal cells to fight Aβ-induced cellular toxicity.

A direct role of ATM in synaptic dysfunctions in AD was hypothesized, considering that ATM directly modulates synaptic function and composition of hippocampal neurons [44,45,47]. Accordingly, in 2019, it was demonstrated that ATM can phosphorylate the developmentally regulated brain protein drebrin (DBN), an F-actin side-binding protein enriched in dendritic spines. Notably, the actin cytoskeleton is a crucial element in maintaining the dendritic spine architecture and in orchestrating the spine’s morphology remodeling driven by synaptic activity [118]. Therefore, an impairment in actin cytoskeleton dynamics can contribute to AD pathology and underlie the synaptic failure in AD. Interestingly, DBN can control spine morphology and function and its progressive loss correlates with cognitive deficits and ageing. Moreover, a noticeable decline in DBN has been found in the hippocampus and cortex of patients with AD [119,120]. Unexpectedly, normal dendritic spine shape and synapse function has been reported in Dbn^−/−^ brains, suggesting that DBN-loss alone is not sufficient to induce synapse dysfunction. Indeed, Kreis et al. demonstrated that an excess of reactive oxygen species (ROS) stimulates ATM-dependent phosphorylation of DBN, enhancing protein stability and improving stress resilience in dendritic spines [86]. Considering that ATM loss has been detected in lysates of specific brain areas of AD patients, the defects in ATM signaling can affect DBN and spine stability, directly participating in synaptic failure in AD.

As well as controlling spine structure, ATM can also be implicated in altering glutamatergic transmission. Importantly, by immunostaining and electrophysiological analyses, synaptic kainate receptors (KARs) have been shown to be impaired in a mouse model of AD, in addition to AMPA and NMDA receptor dysregulation [121]. Of note, synaptic KARs change in *Atm*-deficient neurons and mice through a KCC2-dependent mechanism, supporting a possible contribution of ATM to synaptic dysfunctions in AD.

Overall, these data demonstrate a central role of ATM in the crossing pathways underlying AD pathogenesis. ATM is a messenger of cell death pathways triggered by Aβ but is a kinase that can control spine shape and synaptic transmission as well. Therefore, ATM can be considered a new key synaptic element able to regulate dendritic spine plasticity, required for proper brain function.

**Table 2 cells-12-02181-t002:** Literature review of ATM involvement in neurodegenerative issues, sorted by process.

Process	Paper Title	Year	Ref.
Apoptosis	Requirement for Atm in Ionizing Radiation-Induced Cell Death in the Developing Central Nervous System	1998	[7]
	Defective neurogenesis resulting from DNA ligase IV deficiency requires Atm	2000	[5]
	Reduced NMDA-induced apoptosis in neurons lacking ataxia telangiectasia mutated protein	2003	[115]
	Cell Cycle Activation Linked to Neuronal Cell Death Initiated by DNA Damage	2004	[114]
	Constitutive expression and cytoplasmic compartmentalization of ATM protein in differentiated human neuron-like SH-SY5Y cells	2006	[122]
	DNA-damage response, survival and differentiation in vitro of a human neural stem cell line in relation to ATM expression	2009	[79]
	ATM kinase inhibition in glial cells activates the innate immune response and causes neurodegeneration in Drosophila	2012	[111]
	Functional switching of ATM: sensor of DNA damage in proliferating cells and mediator of Akt survival signal in post-mitotic human neuron-like cells	2012	[123]
	Nuclear accumulation of HDAC4 in ATM deficiency promotes neurodegeneration in Ataxia Telangiectasia	2012	[124]
	EZH2-mediated H3K27 trimethylation mediates neurodegeneration in Ataxia Telangiectasia	2013	[125]
	The Interaction of the Atm Genotype with Inflammation and Oxidative Stress	2014	[126]
	Alteration in 5-hydroxymethylcytosine-mediated epigenetic regulation leads to Purkinje cell vulnerability in ATM deficiency	2015	[80]
	NAD+ replenishment improves lifespan and healthspan in Ataxia telangiectasia models via mitophagy and DNA repair	2016	[127]
	ATM loss disrupts the autophagy-lysosomal pathway	2021	[128]
Autophagy	NAD+ replenishment improves lifespan and healthspan in Ataxia telangiectasia models via mitophagy and DNA repair	2016	[127]
	ATM loss disrupts the autophagy-lysosomal pathway	2021	[128]
Cell cycle	Cell Cycle Activation Linked to Neuronal Cell Death Initiated by DNA Damage	2004	[114]
	Loss of Neuronal Cell Cycle Control in Ataxia Telangiectasia: A Unified Disease Mechanism	2005	[83]
	Oxidative Stress Is Linked to ERK1/2-p16 Signaling-mediated Growth Defect in ATM-deficient Astrocytes	2009	[129]
	Loss of ATM Impairs Proliferation of Neural Stem Cells Through Oxidative Stress-Mediated p38 MAPK Signaling	2009	[130]
	Nuclear accumulation of HDAC4 in ATM deficiency promotes neurodegeneration in Ataxia Telangiectasia	2012	[124]
	EZH2-mediated H3K27 trimethylation mediates neurodegeneration in Ataxia Telangiectasia	2013	[125]
	The Interaction of the Atm Genotype with Inflammation and Oxidative Stress	2014	[126]
	Alteration in 5-hydroxymethylcytosine-mediated epigenetic regulation leads to Purkinje cell vulnerability in ATM deficiency	2015	[80]
	Neurons in Vulnerable Regions of the Alzheimer’s Disease Brain Display Reduced ATM Signaling	2016	[110]
Differentiation	Constitutive expression and cytoplasmic compartmentalization of ATM protein in differentiated human neuron-like SH-SY5Y cells	2006	[122]
	DNA-damage response, survival and differentiation in vitro of a human neural stem cell line in relation to ATM expression	2009	[79]
Double strand breaks (DSBs)	Requirement for Atm in Ionizing Radiation-Induced Cell Death in the Developing Central Nervous System	1998	[7]
	Defective neurogenesis resulting from DNA ligase IV deficiency requires Atm	2000	[5]
	DNA damage induced by polyglutamine-expanded proteins	2003	[131]
	Cell Cycle Activation Linked to Neuronal Cell Death Initiated by DNA Damage	2004	[114]
	DNA-damage response, survival and differentiation in vitro of a human neural stem cell line in relation to ATM expression	2009	[79]
	Early and Late Events Induced by PolyQ-expanded Proteins: IDENTIFICATION OF A COMMON PATHOGENIC PROPERTY OF POLYQ-EXPANDED PROTEINS	2011	[132]
	Functional switching of ATM: sensor of DNA damage in proliferating cells and mediator of Akt survival signal in post-mitotic human neuron-like cells	2012	[123]
	EZH2-mediated H3K27 trimethylation mediates neurodegeneration in Ataxia Telangiectasia	2013	[125]
	Alteration in 5-hydroxymethylcytosine-mediated epigenetic regulation leads to Purkinje cell vulnerability in ATM deficiency	2015	[80]
	NAD+ replenishment improves lifespan and healthspan in Ataxia telangiectasia models via mitophagy and DNA repair	2016	[127]
Epigenetic regulation	Oxidative Stress Is Linked to ERK1/2-p16 Signaling-mediated Growth Defect in ATM-deficient Astrocytes	2009	[129]
	Nuclear accumulation of HDAC4 in ATM deficiency promotes neurodegeneration in Ataxia Telangiectasia	2012	[124]
	EZH2-mediated H3K27 trimethylation mediates neurodegeneration in Ataxia Telangiectasia	2013	[125]
	The Interaction of the Atm Genotype with Inflammation and Oxidative Stress	2014	[126]
	Alteration in 5-hydroxymethylcytosine-mediated epigenetic regulation leads to Purkinje cell vulnerability in ATM deficiency	2015	[80]
	Neurons in Vulnerable Regions of the Alzheimer’s Disease Brain Display Reduced ATM Signaling	2016	[110]
	The impact of glutamine supplementation on the symptoms of Ataxia Telangiectasia: a preclinical assessment	2016	[133]
E/I balance	Reduced NMDA-induced apoptosis in neurons lacking ataxia telangiectasia mutated protein	2003	[115]
Glucose uptake	ATM loss disrupts the autophagy-lysosomal pathway	2021	[128]
Glutamine metabolism	The impact of glutamine supplementation on the symptoms of Ataxia Telangiectasia: a preclinical assessment	2016	[133]
Inflammation/ Immune response	ATM kinase inhibition in glial cells activates the innate immune response and causes neurodegeneration in Drosophila	2012	[111]
	The Interaction of the Atm Genotype with Inflammation and Oxidative Stress	2014	[126]
Insulin signalling	Constitutive expression and cytoplasmic compartmentalization of ATM protein in differentiated human neuron-like SH-SY5Y cells	2006	[122]
	Functional switching of ATM: sensor of DNA damage in proliferating cells and mediator of Akt survival signal in post-mitotic human neuron-like cells	2012	[123]
Lysosomal trafficking & function	ATM loss disrupts the autophagy-lysosomal pathway	2021	[128]
Mitochondrial activity	NAD+ replenishment improves lifespan and healthspan in Ataxia telangiectasia models via mitophagy and DNA repair	2016	[127]
Oxidative stress	DNA damage induced by polyglutamine-expanded proteins	2003	[131]
	Oxidative Stress Is Responsible for Deficient Survival and Dendritogenesis in Purkinje Neurons from Ataxia Telangiectasia Mutated Mutant Mice	2003	[84]
	DNA-damage response, survival and differentiation in vitro of a human neural stem cell line in relation to ATM expression	2009	[79]
	Oxidative Stress Is Linked to ERK1/2-p16 Signaling-mediated Growth Defect in ATM-deficient Astrocytes	2009	[129]
	Loss of ATM Impairs Proliferation of Neural Stem Cells Through Oxidative Stress-Mediated p38 MAPK Signaling	2009	[130]
	ATM is activated by ATP depletion and modulates mitochondrial function through NRF1	2019	[85]
	The Interaction of the Atm Genotype with Inflammation and Oxidative Stress	2014	[126]
	Dramatic extension of tumor latency and correction of neurobehavioral phenotype in Atm-mutant mice with a nitroxide antioxidant	2006	[134]
	Mutation of Ataxia–Telangiectasia Mutated is Associated with Dysfunctional Glutathione Homeostasis in Cerebellar Astroglia	2016	[135]
	NAD+ replenishment improves lifespan and healthspan in Ataxia telangiectasia models via mitophagy and DNA repair	2016	[127]
	ATM phosphorylation of the actin-binding protein drebrin controls oxidation stress-resistance in mammalian neurons and C. elegans	2019	[86]

## 7. Conclusions

In this review, starting from the assumption that AT patients display cognitive and often psychiatric symptoms hardly ascribable to the canonic role of ATM in DNA damage response, we discussed recent and less recent findings related to new meanings of DNA damage formation in neurons and new functions of ATM in neuronal physiology and pathology. Indeed, the evidence that, in response to physiological neuronal activity, DSBs generate in the promoters of stimulus-inducible genes directly leads one to conclude that the correct functioning of DNA damage response machinery is fundamental for proper neuronal function. Also, the excitatory-to-inhibitory switch of GABA, the developmental process occurring at the early postnatal phase necessary in mediating the hyperpolarizing action of GABA neurotransmitters, relies on DNA repair proteins. In particular, genetic studies and results collected with pharmacological inhibitors revealed ATM as a key player in this process. Changes in the GABA switch are at the bases of several neurodevelopmental diseases including autism spectrum disorders and, accordingly, the pharmacological tuning of ATM kinase activity ameliorates electrophysiological defects and cognitive abilities in animal models of autism. Thus, these findings extend the potential investigation of ATM to neurodevelopmental disorders and since, in neurons, DSB repair seems to specifically occur through the activation of the NPAS4–NuA4 complex, the pharmacological targeting of ATM in autism should not affect ability of neurons to repair DNA damages, as they are governed by completely different processes and protein repertoires. 

The selective ability of ATM to control the GABA switch in the hippocampal but not in cortical structure is not surprising. Indeed, increased levels of reactive oxygen species (ROS) have been described in the striatum and cerebellum but not in the cortex of *Atm*^−/−^ mice [136]. Also, reductions in ATM signaling have been highlighted in the frontal cortex of patients with Alzheimer’s disease (AD), whereas increased ATM levels were reported in the cerebellum of AD patients [110]. Finally, robust microglia dysfunctions are evident in the cerebellum but not in the cerebral cortex [6]. Thus, these results emphasize the presence of differences in ATM functions in cellular populations that originate from different brain areas, leading to specific changes in neuronal physiology and pathology. Furthermore, in this context, preclinical studies have described ATM as a messenger of cell death pathways triggered by Aβ and a kinase that can control spine shape and synaptic transmission as well, pointing to ATM as a new key element in AD etiopathogenesis.

Collectively, while several studies have demonstrated the emergent involvement of ATM not only in AT but also in neurodevelopmental and neurodegenerative conditions, a further characterization is necessary to clearly identify (i) new and specific roles of ATM in different brain structures and (ii) consequences linked to its pharmacological inhibition. Results collected so far support researchers in expanding their interests to ATM in new and unexpected disorders.

## Figures and Tables

**Figure 1 cells-12-02181-f001:**
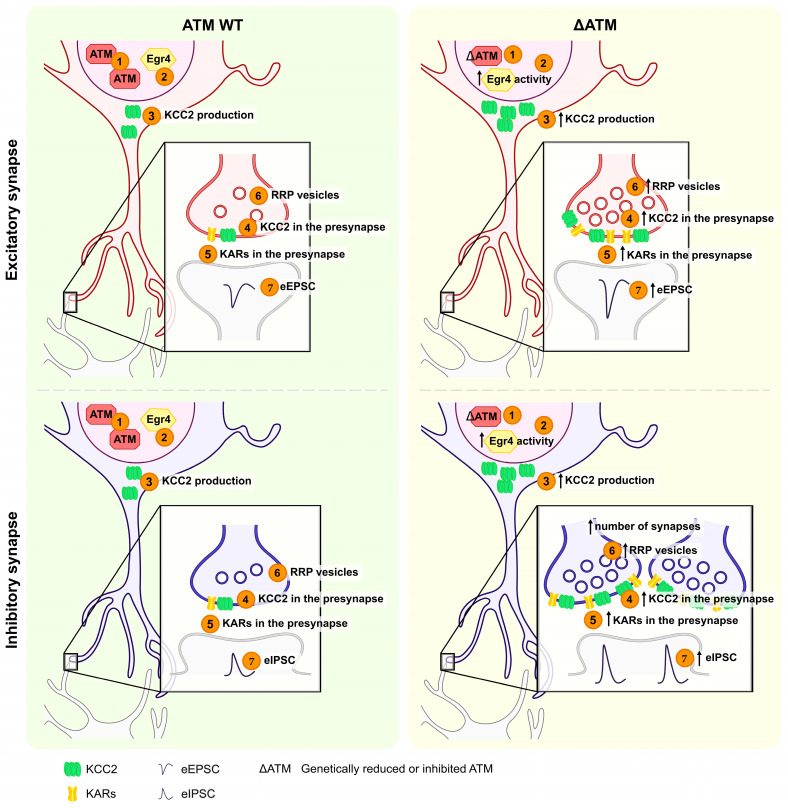
Structural and functional changes in ATM-depleted hippocampal neurons. On the left: excitatory and inhibitory WT neurons with normal (1) ATM expression, (2) Egr4 transcription factor activity, (3–4) KKC2 transcription and protein production, (5) kainate receptor (KAR) levels and (6) excitatory or inhibitory post-synaptic current (EPSC or IPSC) amplitude. On the right: (1) ATM is genetically reduced or pharmacologically inhibited by KU55933 (ΔATM). (2). The rapid Egr4-dependent activation of the *Kcc2b* promoter occurs, leading to (3) enhanced expression of KCC2. Higher KCC2 levels result in a premature development of GABAergic system (GABA-switch) and increased number of GABAergic synapses. (4) KCC2 coexists in a macromolecular complex with synaptic KARs and, at the presynapse, (5) KARs promote (6) clustering of glutamatergic and GABAergic synaptic vesicles in the Readily Releasable Pool (RRP), resulting in (7) higher evoked glutamatergic or GABAergic currents.

## Data Availability

No new data were created for preparing this review.

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
