# Peer review of "New Views of the DNA Repair Protein Ataxia–Telangiectasia Mutated in Central Neurons: Contribution in Synaptic Dysfunctions of Neurodevelopmental and Neurodegenerative Diseases"

_cells, 2023, doi:10.3390/cells12172181_

Round 1

Reviewer 1 Report (Previous Reviewer 1)

The authors have responded very well to the reviewers' suggestions. I support acceptance of this interesting article. 

Author Response

We kindly thank the reviewer for his/her positive response.

Reviewer 2 Report (Previous Reviewer 3)

The authors have made extensive revisions and the current manuscript is much better. However, one minor revision should be made to make the manuscript more scientifically sound. For example, on Page 4 in Line 166, the sentence is not complete. 

The English in this version is much better than the old version.

Author Response

We thank the reviewer for his/her revision to our manuscript. Regarding the comment "on Page 4 in Line 166, the sentence is not complete" actually the sentence is complete but becouse of the marked changes, he/she didn't find the sentence ending (two lines below). We are now uploading a fresh clean copy of the manuscript for an easer reading of the manuscript. 

This manuscript is a resubmission of an earlier submission. The following is a list of the peer review reports and author responses from that submission.

Round 1

Reviewer 1 Report

In this manuscript by Briguglio reviews the function of the DNA damage kinase ATM at the CNS synapse during brain development and degenerative disease. This is a highly exciting topic and this overview promises to provide a valuable overview into a highly relevant and complex topic of research. However, whilst I feel the authors summarise a lot of important work, I find that the review’s organisation and text could be improved. The title suggests that a focus will be placed on discussing specifically non-canonical functions of ATM, which I find is not the case. Therefore, either the title needs to be changed or the article would need to be altered. The structure would clearly benefit from a better set-up, i.e. a more general introduction that clearly introduces ATM and then introduces what will be discussed by this review. Similarly, the end is somewhat abrupt and may benefit from a conclusive paragraph.  The authors seemingly discuss one function of ATM only in the context of one disease state, but never go beyond speculating that a specific function could (very likely) be relevant for another disease state. The English needs to be improved; just one example from the abstract: ‘Being synapses essential units of neuronal physiology responsible for all aspects of cognition, it is coherent that perturbed synaptic functions occur in brain disorders with low cognitive performance’; this sentence needs improvement. The whole review would benefit from careful proofreading, ideally by a native English speaker. Figures that are more numerous as well as more simple may also help comprehending the very compact text. I am recommending acceptance of this interesting work after careful re-organisation of text and improvement of the usage of language.

see above

Reviewer 2 Report

This is a well-written and a timely review paper on the involvement of ATM in synaptic dysfunction and neurodegenerative diseases. While the text reads well, the resolution of Figure 1 can be improved. 

Gene symbols should be capital letters.

The symbol for Ataxia-telangiectasia is (AT) but not (A-T). Please use gene and disease abbreviations according to OMIM. This applies to Ataxia-Telangiectasia Like Disorder (not A-TLD but ATLD1), Fragile X (not FX but FXS) or any other disorder mentioned in the review.

Since this is a review paper, I would add to the references key papers by Tomas Lindahl, Paul Modrich and Aziz Sancar who received the 2015 Nobel prize in Chemistry for mechanistic studies of DNA repair. For example, reference 7 could be a paper from one of the laureates above.

Reference formatting should be checked. I detected on reference where the year is not bold.

Reviewer 3 Report

The authors have thoroughly summarized the contribution of ATM in synaptic dysfunctions of neurodevelopmental and neurodegenerative diseases. However, there are some concerns that need to pay attention.

1. More references regarding the noncanonical function of ATM beyond DNA damage repair should be included in this manuscript too. For example, the following article "Chow HM, et al. ATM is activated by ATP depletion and modulates mitochondrial function through NRF1. J Cell Biol. 2019", which talks about the ATM's role in mitochondria function, should be cited. 

2. As the title is "what's beyond the canonical DNA damage response", the authors should limit the discussion to ATM's canonical role in DNA damage repair. In particular, the first three subsections (1. the DNA damage responses 2. Clinical evidence of DNA repair deficiencies in human brain diseases 3. Role of DNA double-strand breaks (DSBs) formation in neuronal physiology) should be reorganized to make the whole discussion more relevant to the title.

3. The other subsections like 4, 5, and 6 should also be reorganized to make the logic more clear.